# Continuous ultrafiltration during extracorporeal circulation and its effect on lactatemia: A randomized controlled trial

Carlos García-Camacho[1], Antonio-Jesús Marín-Paz[2]*, Carolina Lagares-Franco[3], María-José Abellán-Hervás[4], Ana-María Sáinz-Otero[4]

1 Cardiovascular Surgery Unit, Puerta del Mar University Hospital, Andalusian Health Service, Cadiz, Andalusia, Spain, 2 Nursing and Physiotherapy Department, Faculty of Nursing, University of Cadiz, Algeciras, Spain, 3 Department of Statistics and Operative Research, University of Cadiz, Cadiz, Andalusia, Spain, 4 Nursing and Physiotherapy Department, Faculty of Nursing and Physiotherapy, University of Cadiz, Cadiz, Spain

* antoniojesus.marin@uca.es

**Data Availability Statement:** All statistical data are available from the figshare database (https://

## Abstract

### Introduction

Hyperlactatemia occurs during or after extracorporeal circulation in the form of lactic acidosis, increasing the risk of postoperative complications and the mortality rate. The aim of this study was to evaluate whether continuous high-volume hemofiltration with volume replacement through a polyethersulfone filter during the extracorporeal circulation procedure decreases postoperative lactatemia and its consequences.

### Materials and methods

This was a randomized controlled trial. Patients were randomly divided into two groups of 32: with or without continuous high-volume hemofiltration through a polyethersulfone membrane. Five patients were excluded from each group during the study period. The sociodemographic characteristics, filter effects, and blood lactate levels at different times during the procedure were evaluated. Secondary endpoints were studied, such as the reduction in the intubation time and time spent in ICU.

### Results

Lactatemia measurements performed during the preoperative and intraoperative phases were not significantly different between the two groups. However, the blood lactate levels in the postoperative period and at 24 hours in the intensive care unit showed a significant reduction and a possible clinical benefit in the hemofiltered group. Following extracorporeal circulation, the mean lactate level was higher (difference: 0.77 mmol/L; CI 0.95: 0.01–1.53) in the nonhemofiltered group than in the hemofiltered group (p<0.05). This effect was greater at 24 hours (p = 0.019) in the nonhemofiltered group (difference: 1.06 mmol/L; CI 0.95: 0.18–1.93) than in the hemofiltered group. The reduction of lactatemia is associated with a reduction of inflammatory mediators and intubation time, with an improvement in liver function.

figshare.com/s/919bb3545fd431f0bfc6 | doi:10.
6084/m9.figshare.12115776).

**Funding:** The authors received no specific funding
for this work.

**Competing interests:** The authors have declared
that no competing interests exist.

## Conclusions

The use and control of continuous high-volume hemofiltration through a polyethersulfone membrane during heart-lung surgery could potencially prevent postoperative complications. The reduction of lactatemia implied a reduction in intubation time, a decrease in morbidity and mortality in the intensive care unit and a shorter hospital stay.

## Introduction

Lactate is a biomarker for which its increase or decrease can serve as a predictor of morbidity and mortality in intensive care units (ICUs) [1]. Arterial lactatemia results from the production and elimination of lactate molecules. The concentration of lactate in the body is generally less than 2 mmol/L [2]. When there is a decrease in the supply of oxygen such as anemia or low cardiac output, there is an increase in anaerobic metabolism with the conversion of pyruvate to lactate, increasing its concentration in the blood [3].

Lactate can also be synthesized in critical patients, especially in cases of cardiogenic shock, acute respiratory failure, pneumonia, or sepsis [4, 5]. These pathologies, together with a decrease in clearance due to renal or hepatic failure, contribute to an increase in lactate levels in patients.

Cardiac surgery is relevant due to its relationship with cardiac biochemical processes during extracorporeal circulation (ECC), in which the presence of elevated lactatemia is a predictor of postoperative outcomes [6]. In fact, levels higher than 4.4 mmol/L are related to an increased stay in the ICU and general ward [7].

During ECC, an increase in myocardial and peripheral tissue lactate is demonstrated, associated with impaired tissue oxygenation and early onset hyperlactation, probably due to accelerated anaerobic metabolism as a result of increased circulating epinephrine and inflammatory proteins [3].

Hyperlactatemia is considered when the average blood lactate value exceeds 2 mmol/L [8]. Its occurrence during or after ECC increases postoperative complications, such as infections, while its decrease in the first 24 hours is associated with a decreased mortality rate [9]. However, the occurrence of lactic acidosis during ECC is a complex phenomenon, depending on factors such as duration or hemodilution and ECC time [10, 11]. It is an additional independent risk factor that leads to poor postoperative outcomes. Without the possibility of adequate elimination from the bloodstream, lactate can reach levels above 4 mmol/L, which are associated with an increased risk of postoperative morbidity including a higher rate of 30-day mortality after cardiac surgery [8, 12–15].

Intraoperative lactate measurement is a reliable assessment performed by perfusionists to monitor tissue perfusion during surgical procedures [16]. The objective of the ECC, in addition to eliminating excess liquid, is to eliminate toxic and pro-inflammatory substances. This is a technique that improves hemodynamics, lung function, and hemostasis [16]. Although this combined practice (conventional and modified ultrafiltrations) is considered a safe technique during ECC [17], some authors have associated conventional ultrafiltration with the occurrence of intraoperative hyperlactatemia during the ECC procedure and recommend its use only in situations where the patient suffers from renal failure, a positive fluid balance, poor response to diuretics, or prolonged ECC (more than 120 minutes) [18].

Continuous high-volume hemofiltration with volume replacement is used throughout the ECC procedure to achieve the benefits of both techniques. A polyethersulfone membrane is used for the transfer of solutes by flow dragging and according to the pore size of the membrane to achieve electrolyte and lactate purification. It is important to use solutions with a low

lactate content to fill the circuits of the ECC pump. These compounds have a supraphysiological rate of plasma acetate throughout the ECC process [19, 20], and even small concentrations of acetate produce pro-inflammatory and cardiotoxic effects [21]. However, they are widely used as priming solutions in ECC [22].

Therefore, the objective of our study was to evaluate whether continuous high-volume hemofiltration with volume replacement through a polyethersulfone filter during the ECC procedure decreases postoperative lactatemia and its consequences.

### Primary hypothesis

Continuous high-volume hemofiltration with volume replacement by the use of a polyethersulfone membrane during the ECC procedure in patients undergoing cardiac surgery decreases intraoperative lactatemia.

## Materials and methods

### Design

A randomized controlled trial was conducted between June 2017 and February 2018 at the Puerta del Mar University Hospital (Spain). No variations were made to the trial design or outcomes after trial commencement. This paper uses a trial protocol and the guidelines for reporting parallel group randomized trials (CONSORT); see S1 Protocol and S1 Checklist. The authors confirm that all ongoing and related trials for this drug/intervention are registered.

### Sample size calculation

Consecutive sampling was performed; as patients met the inclusion criteria, they were selected to participate in the study. To determine the sample size, the value of the variance of the response variable was calculated in a reference group [18]. The basic response variable in our study was the lactate elimination rate expressed in amount/unit of time. To determine the sample, we assumed an alpha risk of 0.05, a power of 0.80, a clinically important minimum difference of 0.5, and a standard deviation in the outcome variable of 0.7. The need for a total of 64 participants was finally concluded.

### Ethics

This research conformed to the principles described in the Declaration of Helsinki and was approved prior to its initiation by the ethics committee (IRB) of the Puerta del Mar University Hospital on December 2nd, 2016. All patients who participated in the study signed an informed consent form.

### Participants, recruitment, randomization, and treatment allocation

The inclusion criteria were patients without urgent clinical interventions, patients undergoing extracorporeal circulation normothermic surgical procedures and patients who had a minimum time before decannulation of more than 60 minutes (myocardial reperfusion completed, unclamped aorta, and ECC completed). The exclusion criteria were patients who did not sign the informed consent form, patients with previous renal or hepatic failure, and procedures without ECC. Although some oral antidiabetic agents, such as metformin, may alter lactate levels [23], patients with diabetes were not excluded from the study because the preoperative lactate values of these patients were within the limits of normality.

For the patient recruitment, a previous interview was conducted 24 hours before the surgery after the hospital admission; in this interview the patients were informed about the

components of extracorporeal circulation, the hemofiltration technique, and the study overview; they were also informed that, although they had signed the informed consent, they could refuse to participate in the study at any time. Recruitment began on September 1st, 2017 and ended on February 28th, 2018.

The study was blinded to the patients, data analysts, and ICU staff. The allocator randomly divided the patients into a control group (CG) or a hemofiltered group (HG). In the HG a polyethersulfone filter was used throughout the ECC while in the CG conventional procedures without hemofiltration were used. Surgeries were randomly assigned to four groups: 44 cases of valvular surgery (68.75%), 9 cases of coronary surgery (14.06%), 6 cases of valvular and coronary surgery (9.38%), and 5 cases of aortic and ascending aortic replacement surgery (with the Bentall technique) (7.81%).

The allocator was assigned by the head of the hospital's ethics committee. He randomly assigned patients into eight blocks, all being equivalent in all procedures except in the treatment maneuvers; there were no notable differences between the possible confounders measured in the two analyzed groups. There was no stratification.

## Procedure

In the present study, all patients were operated on under propofol-induced general anesthesia and maintained during the procedure with the volatile anesthetic agent sevoflurane, including during the ECC period. All procedures were performed through a median sternotomy, and normothermia (309.15 K; 36°C; 96.8°F) was maintained in all patients throughout the procedure.

An ECC open circuit consisting of a set of polyvinyl chloride tubes and a polypropylene membrane oxygenator with an integrated arterial filter with a coating based on phosphorylcholine molecules was used. The ECC device consisted of a biopump, and all procedures were performed with a centrifugal pump. In HG, the hemofiltration membrane used was a membrane made of polyethersulfone, with a surface area of 1.35 $m^2$.

The priming solution used contained (1) 80 mg dexamethasone, (2) 50 mg ranitidine, (3) 150 IU/kg sodium heparin, (4) 500 mL of a solution containing 40 mg/mL succinylated gelatine, and (5) 900 mL comprising 2 mmol/L of $K^+$, 32 mmol/L of $HCO^{3-}$, 111.5 mmol/L of $Cl^-$, 140 mmol/L of $Na^+$, 0.5 mmol/L of $Mg^{2+}$, 1.75 mmol/L of $Ca^{2+}$, 6.1 mmol/L of glucose, 3 mmol/L of lactate, and 250 mL of mannitol 20%.

The cardioplegia solution and procedure used in the surgical procedure was a modification of that developed by Calafiore et al. [24], and the cardioplegia solution was administered antegrade through the aortic artery and retrograde through the coronary sinus. For this purpose, 80 mEq of KCl and 1.5 grams of $MgSO_4$ were mixed in a 50 $cm^3$ syringe using a volumetric infusion pump connected to the circuit through a three-step key.

The data were collected with a CONNECT® system (LivaNova Deutschland, Münich, Germany). To measure lactate levels, a GEM premier 4000® analyser was used, with amperometric biosensors connected to the ECC pump and a recording system for further statistical analysis [25, 26]. The blood flowed from the oxygenator to the hemofilter through a recirculation line with a flow of 100 to 500 mL/min, depending on the time of surgery, without exceeding the maximum transmembrane pressure of 500 mmHg as recommended by the manufacturer. The effluent rate was 110 ml/min to an average of 80 ml/kg/h, similar to HERO-ICS study [27].

The HG was subjected to high-volume hemofiltration together with replacement of the hemofiltered liquid with a solution used in extrarenal purification techniques comprising 2 mmol/L of $K^+$, 32 mmol/L of $HCO^{3-}$, 111. 5 mmol/L of $Cl^-$, 140 mmol/L of $Na^+$, 0.5 mmol/L of

$Mg^{2+}$, 1.75 mmol/L of $Ca^{2+}$, 6.1 mmol/L of glucose, and 3 mmol/L of lactate. Pursuing a zero balance, this solution allows for the replacement of 3,000 $cm^3$ per hour with a maximum of 60,000 $cm^3$/day.

## Data analysis

The studied variables included biometric and analytical data that were analyzed preoperatively (age, sex, height, weight, and EuroSCORE), intraoperatively (diuresis, time of clamping, time of ECC, and attendance time between unclamping of the aorta and the completion of ECC), and post-operatively at 24 hours after surgery or thereafter (time of use of inotropic agents, C-reactive protein, intubation time, and time spent in ICU). Lactate and hemoglobin levels were measured at all stages up to 24 hours after surgery. In the HG group, ultrafiltration was performed from the beginning of ECC to the end. Blood samples were collected prior to initiation of ECC; every 20 minutes during the procedure, with the highest value recorded; at the end of the ECC; and 24 hours after admission to the ICU.

The units of the variables were: Age (years), sex (male/female), height (cm), weight (kg), EuroSCORE (1–6 points), diuresis ($cm^3$), time of clamping (minutes), time of ECC (minutes), attendance time between unclamping of the aorta and the completion of ECC (minutes), lactate (mmol/L), hemoglobin (g/dL), time of use of inotropic agents (hours), C-reactive protein (mg/L), intubation time (hours), and time spent in ICU (days).

The parameters of oxygen supply, oxygen consumption, venous oxygen saturation and oxygen extraction were continuously recorded throughout the procedure [28]. After hemodynamic and respiratory stability with effective cough and neurological stability, spontaneous T-tube ventilation was initiated. If the patient's stability persisted, extubation was performed [29]. These variables and procedures have allowed us to study secondary endpoints, including possible benefits in patients with renal failure and the reduction in the intubation time, time spent in ICU and C-reactive protein levels.

Quantitative variables were expressed by arithmetic means and standard deviations, or medians and interquartile ranges. Qualitative variables were expressed as frequencies and percentages. The normality of continuous variables was evaluated with the Kolmogorov-Smirnov test. For the analysis of intergroup changes, the following analyses were performed: ANOVA-RM test with a post hoc least significant difference (LSD) test, Student's t-test to evaluate the difference between the means of two independent groups, and the Mann-Whitney U test for nonnormally distributed variables. To evaluate the statistical independence of the categorical variables, the chi-square test was used. Statistical significance was considered at $p < 0.05$.

## Results

Of the 64 patients enrolled in this study, 32 were randomly assigned to each group (CG and HG). During the assignment process, the following patients were excluded from the CG: 1 due to the need for ventricular assistance at the end of ECC [30] and 4 for hemofiltration due to preoperative anemia. In the HG, 3 of the patients were excluded from the study due to bleeding that resulted in a new intervention within 24 hours after surgery, 1 was excluded due to mediastinitis requiring a prolonged stay in the ICU, and 1 was excluded due to death from intraoperative vasoplegia without response to vasoactive drugs. Fig 1 demonstrates the flow of participants throughout the trial.

### Participant characteristics

Table 1 shows the descriptive results of the sociodemographic, anthropometric and biochemical variables analyzed in both groups. With respect to the sex of the patients studied, there

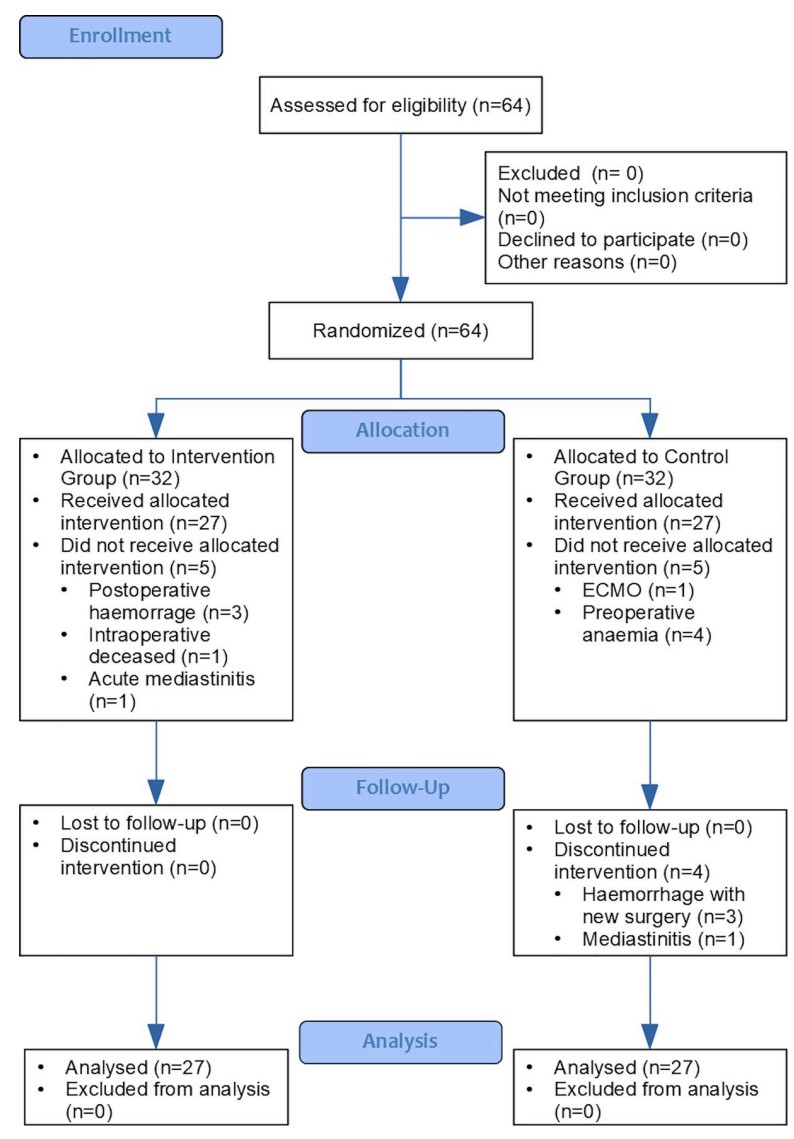

**Fig 1. Study flow chart (CONSORT).**

**Table 1. Comparative analysis of sex, age, height, weight, and serum lactate prior to the initiation of ECC between the HG and CG.**

| Variables | Total (N = 54) | Group | |
|---|---|---|---|
| | | CG (n = 27) | HG (n = 27) |
| Sex: | | | |
| Male | 66.7% (36) | 77.8% (21) | 55.6% (15) |
| Female | 33.3% (18) | 22.2% (6) | 44.4% (12) |
| Age (years) | 63.30 (11.1) | 62.81 (11.6) | 63.78 (10.8) |
| Height (cm) | 165.59 (10.5) | 167.41 (8.7) | 163.78 (11.8) |
| Weight (kg) | 79.35 (15.4) | 79.13 (15.2) | 79.57 (15.9) |

were 36 (66.7%) men and 18 (33.3%) women. The mean age of the total sample was 63.3 years (CI: 0.95: 60.3–66.3; SD 11.1).

## Patients' hemodynamics

Previous average hemoglobin was 12.43 g/dL (CI: 0.95: 11.93–12.92; SD 1.82). Levels dropped to a minimum of 8.28 g/dL during ECC (CI: 0.95: 7.88–8.67; SD 1.45), rising to 10.49 g/dL after ECC (CI: 0.95: 7.35–13.63; SD 11.50) and averaging 10.70 g/dL (CI: 0.95: 10.27–11.12; SD 1.56) 24 hours after admission to the ICU. There were no significant differences between HG and CG in all hemoglobin measurements collected during the study.

In the total of the participants, the time of use of inotropics has been distributed in the range between 0 and 336 hours, with a median of 24 hours. The mean time is 45.85 h (CI: 0.95: 27.22–64.48; SD 68.26). Dividing the sample into the groups according to the hemofiltrate, it is observed that the use of inotropic drugs is not normally distributed (p<0.000 in Kolmogorov-Smirnov test). The mean value of cases with filter (49.85 hours) is somewhat higher than the mean value of patients who have not had filter (41.85 hours). But this difference is not statistically significant ($Z_U = 0.04$; p = 0.972).

By segmenting according to the EuroSCORE risk grades of the patients, the results indicate that in the group of low risk patients, the difference in the time of use of inotropics is high. Therefore, in this case, even having also high variability in the CG and the reduced number of low risk patients in the EuroSCORE, the difference becomes statistically significant (p = 0.02). Furthermore, the size of the effect (very large: 51.5%) supports the existence of this relationship, according to which the time of use of inotropics is higher in the CG (Table 2). In addition, attendance time between unclamping of the aorta and the completion of ECC is less variable and is reduced in low-risk EuroSCORE HG patients, so hemodynamic parameters remain more stable than in the same EuroSCORE in CG ($Z_U = 2.32$; p = 0.02). There are no significant differences in the rest of the patients with medium and high EuroSCORE risk about variables time of use of inotropic agents and attendance time between unclamping of the aorta and the completion of ECC.

## Lactate levels

Lactate measurements were performed by perfusionists at baseline, during ECC (the maximum value was considered), post-ECC, and at 24 hours in each group. In both groups, there was a clear elevation in the mean values obtained during and after ECC as well as at 24 hours with respect to the reference values. The differences between the measurements were significant (p<0.001), with an effect size of 30.6% in the CG and 37.1% in the HG.

When the initial lactate measurement was excluded and the comparison was made with only the latter three measurements (maximum during ECC, post-ECC, and at 24 hours), the analysis showed two results. First, the CG retained statistical significance, but the effect size was reduced to 12.7%, which is moderate-high. Second, there were no significant overall differences in the HG (mild effect of 4.1%). As such, in both groups, these results clearly

**Table 2. Intergroup inferential analysis.** Comparison of the averages of the Inotropic Time (hours) between CG and HG and according to the EuroSCORE.

| EuroSCORE | CG (n = 27) | | HG (n = 27) | | t-test | | $R^2$ |
|---|---|---|---|---|---|---|---|
| | n | Mean (SD) | n | Mean (SD) | Value | p | |
| Low risk group (n = 8) | 4 | 94.50 (72.40) | 4 | 3.00 (2.45) | $Z_U = 2.32$ | 0.020 | 0.515 |
| Medium risk group (n = 23) | 10 | 25.10 (44.62) | 13 | 50.23 (88.32) | $Z_U = 1.33$ | 0.184 | 0.031 |
| High risk group (n = 23) | 13 | 42.23 (40.58) | 10 | 63.30 (91.50) | $Z_U = 0.16$ | 0.876 | 0.026 |

**Table 3. Intragroup variation in the lactate values according to the time of measurement.** Contrasts between pairs of measurements (LSD test) and overall contrast (ANOVA-MR).

| Measurement | CG (n = 27) | | | | | HG (n = 27) | | | | |
|---|---|---|---|---|---|---|---|---|---|---|
| | Mean (SD) | Previous[d] | Max. ECC[e] | Post-ECC[f] | 24 h[g] | Mean (SD) | Previous[d] | Max. ECC[e] | Post-ECC[f] | 24 h[g] |
| Previous[d] | 1.01 (0.47) | — | 0.000 | 0.000 | 0.000 | 0.84 (0.27) | — | 0.000 | 0.000 | 0.000 |
| Max. during ECC[e] | 1.86 (1.06) | 4.50[c] | — | 0.007 | 0.011 | 1.65 (0.45) | 10.50[c] | — | 0.207 | 0.080 |
| Post-ECC[f] | 2.70 (1.57) | 5.54[c] | 2.30[c] | — | 0.646 | 1.93 (1.18) | 5.04[c] | 1.30 | — | 0.872 |
| 24 h | 2.95 (2.12) | 4.82[c] | 2.72[b] | 0.46 | — | 1.89 (0.80) | 7.04[c] | 1.82 | 0.16 | — |
| | ANOVA-RM[a] | $F = 11.46$; $p = 0.000$; $R^2 = 0.306$ | | | | ANOVA-RM[a] | $F = 15.33$; $p = 0.000$; $R^2 = 0.371$ | | | |
| | Linear test | $F = 32.87$; $p = 0.000$; $R^2 = 0.558$ | | | | Linear test | $F = 49.29$; $p = 0.000$; $R^2 = 0.655$ | | | |
| | Quadratic test | $F = 1.11$; $p = 0.301$; $R^2 = 0.041$ | | | | Quadratic test | $F = 11.62$; $p = 0.002$; $R^2 = 0.309$ | | | |

[a]Repeated measures ANOVA.

[b]Significant difference at 5% ($p<0.05$).

[c]Highly significant difference at 1% ($p<0.01$).

[d]Preoperative level.

[e]Maximum value in ECC.

[f]Immediately after ECC.

[g]After 24 hours of ECC.

demonstrated that the differences between the previous value and the latter three were significant ($p<0.001$), with the previous value being lower than all the others. Likewise, in the HG the values of these three measures did not differ significantly from each other, whereas in the CG lactate increased significantly from the maximum ECC value to the post ECC ($p<0.01$) and 24-hour values (only $p<0.05$ due to the high variability in this measurement). Between these last two measurements (after ECC and 24 hours), the difference was no longer significant (Table 3).

Regarding the analysis between groups (Table 4), the results show that there is also no statistically significant difference between the mean values of maximum lactate in ECC. However, from there on, significant differences already appear. At the post-ECC moment, the mean lactate value is higher (difference: 0.77 mmol/L; IC 0.95: 0.01–1.53) in CG ($p = 0,047$) although with moderate effect (7.4%). The 24-hour mean lactate value is even higher ($p = 0.019$) in CG (difference: 1.06 mmol/L; IC 0.95: 0.18–1.93).

Finally, an intergroup analysis was performed on different parameters. Statistical significance was found for intubation time, and there was a noticeable difference in the length of stay in the ICU between the two groups. The length of stay in the ICU was lower in the HG than in the CG.

**Table 4. Intergroup inferential analysis.** Comparison of lactate means between CG and HG, at each measurement time.

| Time | CG (n = 27) | HG (n = 27) | t-test | | $R^2$ |
|---|---|---|---|---|---|
| | Mean (SD) | Mean (SD) | Value | p | |
| Previous[a] | 1.01 (0.47) | 0.84 (0.27) | 1.59 | 0.117 | 0.047 |
| Max. during ECC[a] | 1.86 (1.06) | 1.65 (0.45) | 0.94 | 0.349 | 0.017 |
| Post-ECC[b] | 2.70 (1.57) | 1.93 (1.18) | 2.04 | 0.047 | 0.074 |
| 24 h | 2.95 (2.12) | 1.89 (0.80) | 2.43 | 0.019 | 0.102 |

[a]Preoperative level.

[b]Maximum value in ECC.

[c]Immediately after ECC.

**Table 5. Intergroup inferential analysis.** Effect of filter use on different parameters.

| Variables | Total (N = 54) | CG (n = 27) | HG (n = 27) | Contrast test | | $R^2$ |
|---|---|---|---|---|---|---|
| | | | | Value | p | |
| Time of ECC (minutes) | 99.87 (44.82) | 104.11 (51.94) | 95.63 (36.87) | t = 0.69 | 0.492 | 0.009 |
| Blood transfusion (Yes) | 22.2% (12) | 22.2% (6) | 22.2% (6) | $Chi^2$ = 0.00 | 1 | ———— |
| Intubation time (hours) | 6.48 (4.96) | 7.93 (6.38) | 5.04 (2.30) | $Z_U$ = 2.29 | 0.022 | 0.086 |
| Time spent in ICU (days) | 4.04 (2.59) | 4.11 (1.74) | 3.96 (3.26) | $Z_U$ = 1.82 | 0.069 | 0.001 |
| Renal failure (Yes) | 9.3% (5) | 14.8% (4) | 3.7% (1) | $Chi^2$ = 1.98 | 0.159 | 0.040 |
| C-reactive protein (mg/L) | 81.69 (37.21) | 85.08 (42.03) | 78.30 (32.13) | t = 0.67 | 0.509 | 0.008 |

The relationship between this time factor and the lactate levels was not the same in each group. In the CG, the only possible relationship between these two variables was a direct linear relationship ($p < 0.001$; $R^2$ = 55.8%). In the HG, although the most likely relationship was also linear ($p < 0.001$; $R^2$ = 65.5%), the decrease in the last measurement indicated a quadratic-type association ($p < 0.01$; $R^2$ = 30.9%).

The mean values of the arms are similar with a small difference of 8.5 minutes. The time in CG being higher, although the difference does not reach statistical significance. There is no difference in the use of blood transfusions between the two groups.

The intubation time variable is not normally distributed ($p < 0.001$ in the Kolmogorov-Smirnov test) due to a large asymmetry with accumulation of cases in the low values versus very few with high values. The total mean is 6.48 (CI 0.95: 5.13–7.84) with a median of 5 in a range of 1 to 31 hours. Using the Mann-Whitney U-test, statistically significant differences were found ($p < 0.05$) so that according to the data of the averages (both mean and median) the time is somewhat higher (about 3 hours) in CG. The effect size equivalent to this significance is moderate (8.6%).

The time spent in ICU is also not normally distributed ($p < 0.01$ in the Kolmogorov-Smirnov test) due to the concentration of cases in the low values as in the previous one. The mean time of the total group is 4 days (CI 0.95: 3.33–4.74) and the mean values (mean and median) of both groups are very similar to each other.

There is no statistical significance between groups in renal failure. Actually, the number of cases with renal failure is very small, that in spite of the observed difference, this result must be taken with caution and is not sufficient statistical evidence to be able to intuit an effect.

C-reactive protein is normally distributed with an average value of 81.69 mg/dL (IC 0.95: 71.53–91.85). The mean value of CG is slightly higher than the mean value of HG but this difference does not reach the statistical significance so our data does not allow us to admit that the filter use factor is statistically related to C-reactive protein values (Table 5).

## Discussion

Several authors have identified preoperative factors that favor the onset of hyperlactatemia during ECC [13]. Patients with type II diabetes mellitus often increases lactate associated with reduced aerobic oxidative capacity and restricted lactate transport. There are also factors that associate the increase in lactate concentration with age, sex, and comorbidities. Anemia is another factor associated with hyperlactatemia by decreasing oxygen supply and producing tissue hypoxia even with normal intravascular volume [31, 32].

However, there are other mechanisms responsible for the appearance of hyperlactatemia in the intraoperative period of cardiac surgery with ECC: (1) the duration of surgery with ECC, with a relationship directly proportional to the time of surgery [10, 33], and (2) a deficit in

oxygen intake and consumption as well as an increase in its extraction [28] as they affect the morbidity of these patients, especially when lactate levels are higher than 4 mmol/L [7].

Nonetheless, the results obtained show that high-volume hemofiltration pursues a zero balance at the end of ECC, managing to mitigate the presence of high amounts of lactate regardless of sex and previous pathologies in patients during the preoperative period as well as the duration of ECC.

In the context of this background, it was important to assess the difference between the two studied groups; in the preoperative period, the elevation of lactate in the HG from ECC until the last analysis at 24 hours had no significance (p>0.05). However, there was a significant increase in the CG (p<0.01), which was independent of the time of ECC.

We must emphasize that in our study at all temporal points the optimal values of oxygen contribution, consumption, and extraction were maintained because gases and the lactate levels were measured to control pH and oxygenation. Even so, hyperlactatemia appeared during and after ECC, probably due to alterations in microcirculation [3]. However, the level of hyperlactatemia was lower in the HG than in the CG, contrary to what Soliman et al. reported [18]; this is possibly attributable to the low concentration of lactate in the extracorporeal circuit priming solution used in this study (3 mmol/L) compared with that in Ringer's solution (27 mmol/L).

It is important to highlight the time period from ECC departure until 24 hours after surgery. In the CG, lactate continued to rise; however, in the HG, the reverse occurred. This quadratic association can be attributed to a reduced need for clearance of lactate by the kidneys and liver, thus improving liver function [34, 35]. This theory is also supported by the fact that in postoperative patients who underwent cardiac surgery with ECC, increased lactate in the absence of dysoxia can be caused by an exacerbated inflammatory response, mitigated by the use of continuous high-volume hemofiltration with a polyethersulfone filter [36], thus resulting in a zero balance at the end of the procedure [37].

The results obtained in this study showed a decreased lactate level in the HG during ECC, at the end of ECC, and 24 hours after surgery. The HG did not show significant changes in contrast to the CG, possibly due to the permeability of the polyethersulfone membrane. This membrane has a lactate screening coefficient equal to 1; thus, the concentration of lactate obtained in the effluent is equal to that existing in the plasma.

Lactate elevation in the intra- and postoperative periods is due to complex mechanisms rather than a single cause [38, 39], just as not all patients develop it in the same way. A number of mechanisms can be involved during ECC, such as low oxygen supply [40] and prolonged ECC time [41]. Currently, lactate deficit clearance in postoperative patients has been demonstrated to be an independent risk factor for poor outcomes in postoperative cardiac surgery with ECC patients [42].

On the other hand, lactate levels greater than 3 mmol at 6 hours after surgery increase the probability of major complications [14, 43]. Therefore, these results suggest prove that in both group, lactate varied significantly depending on the condition/time at which the measurement was performed.

As for secondary outcomes, the use of inotropic agents in the postoperative period in relation to EuroSCORE indicates that low risk patients (0–2 points) benefit more from the technique of continuous hemofiltration with volume replacement, as non-hemofiltrators require a longer time of use of inotropes. In fact, according to a recent study, the increased time of use of inotropes implies an increased risk of morbidity and mortality after cardiac surgery [44]. In the same way, the study shows that there is an improvement in the hemodynamic stability of hemofiltrated patients at low risk in the EuroSCORE due to a shorter attendance time. However, these results should be interpreted with caution due to the low number of patients included in this category of the EuroSCORE.

Lactate reduction decreases mechanical ventilation time of patients undergoing cardiac surgery with ECC. The length of stay in the ICU was also significantly reduced as a postoperative intubation time greater than 12 hours is directly related to an increased length of stay [45]. Moreover, it prevents the appearance of high lactate levels caused by long stays in the ICU [46, 47]. This result is due to the use of hemofiltration and extracorporeal circuit priming solution [48].

The use of intraoperative hemofiltration may be beneficial to the patient in the short term as well as to patients with preoperative renal dysfunction in the long term [49]. Additionally, it may also be recommended for patients with previous liver disease [36].

## Strengths and limitations

In some countries, perfusionists collaborate with surgeons and anesthetists to control and maintain ECC in patients before, during, and after surgery. The findings of this study involve a series of interventions in clinical perfusionist practice to eliminate the increase in serum lactate in surgical interventions with ECC and thereby reduce intubation time and morbidity and mortality in the ICU as well as improved liver function.

The main limitation of this study was a lack of previous research on continuous high-volume hemofiltration with volume replenishment during ECC, preventing proper comparisons with other studies. Moreover, more research focusing on these results with an equal or greater sample of participants in this study is needed to obtain more consistent results.

Other limitations were the variability in the surgical procedures, the interventions by different surgeons during the procedure that could have affected the time of surgery, the time of ECC and the different medical and nursing teams during the postoperative period in the ICU [11]. In addition, there are currently no analytical determinations that can discern the type of hyperlactatemia that patients develop. However, this study aimed to reduce hyperlactatemia regardless of its origin.

## Conclusions

Although the occurrence of hyperlactatemia is common at the end of the ECC procedure, protocols aimed at reducing intra- and postoperative lactate levels through continuous high-volume hemofiltration with volume replacement and a zero balance favor the elimination of lactate during the ECC procedure. This is reflected in the lactate levels at the end of ECC and at 24 hours. The serum lactate levels in patients after ECC are decreased when continuous high-volume hemofiltration with a polyethersulfone membrane is used. The reduction in lactate levels within 24 hours after ECC in the CG was related to a decreased concentration of serum lactate after ECC, allowing improved purification. Moreover, these results could potentially stabilize the hemodynamics in low cardiac risk patients. This study could be fundamental to establishing specific protocols for its use in cardiac surgery with ECC, which, in combination with postoperative nursing care, could shorten the duration of care of individuals undergoing this type of intervention.

## Supporting information

**S1 Checklist. CONSORT checklist.**
(DOC)

**S1 Protocol. Trial protocol.**
(DOC)

**S2 Protocol.**
(DOC)

## Author Contributions

**Conceptualization:** Carlos García-Camacho, Antonio-Jesús Marín-Paz.

**Data curation:** Carlos García-Camacho.

**Formal analysis:** Carolina Lagares-Franco.

**Investigation:** Carlos García-Camacho.

**Methodology:** María-José Abellán-Hervás, Ana-María Sáinz-Otero.

**Project administration:** María-José Abellán-Hervás, Ana-María Sáinz-Otero.

**Resources:** Carlos García-Camacho.

**Validation:** Antonio-Jesús Marín-Paz.

**Visualization:** Antonio-Jesús Marín-Paz.

**Writing – original draft:** Carlos García-Camacho, Antonio-Jesús Marín-Paz.

**Writing – review & editing:** Antonio-Jesús Marín-Paz, Carolina Lagares-Franco, María-José Abellán-Hervás, Ana-María Sáinz-Otero.

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
