## [Decision Letter · Decision Letter 0]

15 Jul 2020

PONE-D-20-10699

Continuous ultrafiltration during extracorporeal circulation and its effect on lactatemia: A randomized controlled trial

PLOS ONE

Dear Dr. Marín-Paz,

Thank you for submitting your manuscript to PLOS ONE. After careful consideration, we feel that it has merit but does not fully meet PLOS ONE’s publication criteria as it currently stands. Therefore, we invite you to submit a revised version of the manuscript that addresses the points raised during the review process.

As you will gather from their comments, the referees raised several important issues (see below) that deserve specific attention. Therefore, a major revision is required before we can consider the manuscript for publication in PLOS ONE. Please read the entire contents of this letter and address carefully all of the comments and the reviewers' critiques. Please note that the paper cannot be accepted until you have addressed all the reviewers' critiques. Depending upon the editors' evaluation of the revision, your paper may or may not have to go back to the reviewers.

We look forward to receiving your revised manuscript.

Kind regards,

Gianpaolo Reboldi, MD, MSc, PhD

Academic Editor

PLOS ONE

Journal Requirements:

2. Please ensure that you include a title page within your main document.

We do appreciate that you have a title page document uploaded as a separate file, however, as per our author guidelines (http://journals.plos.org/plosone/s/submission-guidelines#loc-title-page) we do require this to be part of the manuscript file itself and not uploaded separately.

3. Thank you for submitting your clinical trial to PLOS ONE and for providing the name of the registry and the registration number. The information in the registry entry suggests that your trial was registered after patient recruitment began. PLOS ONE strongly encourages authors to register all trials before recruiting the first participant in a study.

i) your reasons for your delay in registering this study (after enrolment of participants started);

ii) confirmation that all related trials are registered by stating: “The authors confirm that all ongoing and related trials for this drug/intervention are registered”.

Please also ensure you report the date at which the ethics committee approved the study as well as the complete date range for patient recruitment and follow-up in the Methods section of your manuscript.

Reviewers' comments:

Reviewer's Responses to Questions

**Comments to the Author**

1. Is the manuscript technically sound, and do the data support the conclusions?

Reviewer #1: Partly

Reviewer #2: Partly

Reviewer #3: Yes

2. Has the statistical analysis been performed appropriately and rigorously? 

Reviewer #1: No

Reviewer #2: Yes

Reviewer #3: Yes

3. Have the authors made all data underlying the findings in their manuscript fully available?

Reviewer #1: No

Reviewer #2: Yes

Reviewer #3: Yes

4. Is the manuscript presented in an intelligible fashion and written in standard English?

Reviewer #1: No

Reviewer #2: Yes

Reviewer #3: Yes

5. Review Comments to the Author

Reviewer #1: this paper looks at a randomised controlled trial of 64 patients of whom 54 contributed data.

The English needs some minor attention throughout (eg. potency should be power, and I think variables of confusion may actually mean confounders).

Given the fact that 64 patients with data were required why was no margin for error built in - there would appear to be about a 15% dropout; it would also appear that these dropouts may be informative given that they are associated with clinical outcome. How was this allowed for in the analysis? Why are no aptietns ewcluded from analysis in the CONSORT diagram - 10 are excluded from a full ITT analysis.

Patients were randomised into one of 2 arms, not 8 blocks - do you mean the block size in the randomisation was 8? Was there any stratification?

It is statistically incorrect to test baseline characteristics by arm. Units are missing from table 2.

The numbers male and female are wrong as the HG numbers add up to 29 not 27

It is incorrect to analyse the primary outcome within arm - please perform a proper repeated measures analysis adjusted for baseline. As such the contrast between groups and the CI are not given here- I also assume the GC really means CG.

The data require a proper analysis to give the treatment effect and CI with inferences based upon the CI. Until these are done the value of the intervention is not assessable.

Table 4 uses intervention and cnotrl - the rest of the paper uses HG and CG.

Reviewer #2: Dear Authors,

I have read with interest your paper entitled "Continuous ultrafiltration during extracorporeal circulation and its effect on lactatemia: A randomized controlled trial".

In my opinion, the biggest flaw is that the main outcome of this trial was the “only” reduction of lactatemia, which it may be interesting per sè, but you should provide further insights to better interpret the findings of the study. Accordingly, you declare in the abstract that the aim of study is “whether continuous ultrafiltration…. decreases post-operative lactatemia and its consequences”.

What are these consequences? You reported the reduction of lactatemia in methods (in the abstract and in the main body) as unique outcome. In the phase of the trial design did you think other secondary outcomes in addition to reduction of blood lactate levels?

Moreover, how do you interpret the reduction of lactatemia? Is it a result of better depuration or a consequence of the general improvement of hemodynamic conditions and/or inflammation? Can you provide further insight to clarify this point?

Furthermore, there is a formal issue, because style of this manuscript is similar to a narrative review (mostly in the introduction and discussion sections). It needs a further effort to improve this point.

Following my point-by-point considerations:

- Abstract:

o Please clarify the points described above, changing it accordingly.

- Introduction section:

o Please reduce this section, providing only the information tightly related to the topic of the manuscript (e.g.: please

cancel the paragraph about use of type of solutions, etc.).

- Methods section:

o Please cancel table 1 and move the information from the table to text.

o Please add secondary endpoints, if it is possible.

- Results section:

o Please add information about patients’ hemodynamics and other vital parameters for improving the interpretation of findings.

o Please add in the table 3 the value at baseline and the intragroup differences (pre- and – post continuous UF).

o Please use regression analysis to analyze the effect after correction for confounders (ECC time, etc.)

o Please clarify the legend and the parameters included in the table 4.

- Discussion section:

o Depending on these additional results, please reconsider the discussion section and the limitations of the study.

o Please rewrite discussion, highlighting the main findings of study and focused on the interpretation of these findings.

I hope that my comments may be helpful for improving your manuscript.

Kind regards

Reviewer #3: In this randomized controlled trial, the Authors have explored the effects of high-volume hemofiltration with a polyethersulfone membrane versus conventional treatment on postoperative lactate levels in patients undergoing extracorporeal circulation normothermic surgical procedure. Although there are multiple studies on the role of high-volume hemofiltration in critically ill patients with AKI and sepsis, the number of randomised trials on high-volume hemofiltration post-cardiac surgery is quite limited, so additional evidence is very welcome. The paper is well written and scientifically sound and has been appropriately reported according to the CONSORT checklist. Hence, the paper might deserve publication on Plos One, provided that the Authors can satisfactorily address some comments, as detailed below:

1) Although I do appreciate that the topic is complex and requires some background information, the introduction is probably a bit too long and could be slightly shortened

2) As there is no universal consensus on the definition of high-volume hemofiltration, the Authors should provide additional information on the rate of hemofiltration. Based on the results of two large randomized-controlled trials (N Engl J Med 2009, 361:1627–1638; N Engl J Med 2008, 359:7–20) and subsequent systematic reviews (Crit Care Med 2010, 38:1360–1369;Clin J Am Soc Nephrol 2010, 5:956–963)3, the adequate dose of hemofiltration treatment for acute kidney injury ) has been defined as an effluent rate between 25 and 30 ml/kg/hour, which is consistent

with HVHF being defined by an effluent rate exceeding 35 ml/kg/hour. How was the hemofiltration rate in this trial? On page 9, line 11 the Authors have mentioned some information on the blood flow (between 100 and 500 mL/min depending on the time of the surgery), but I could not find any information on the average effluent rate. Please provide this information.

3) Page 6, design. Please specify the name of the hospital in Andalusia. Similarly, in the ethics section on page 7, please provide the name of the hospital where the ethics committee was based.

4) Page 7, line 3 from bottom. Please provide additional details on the "hemophilia technique". Was this a typo for hemofiltration?

5) Page 8, line 1. Please provide additional details on the allocator and the randomization technique. Who was the allocator? Was he one of the Authors or an external allocator? How were the patient assigned to the eight different blocks?

6) Page 8, line 7 from bottom. Please provide the full definition of the acronym "PVC"

7) Page 9, line 9. Please provide a supporting reference got the analyzer which was used in the study (including the name of the manifacturer)

8) Page 9, line 2 from bottom. Please provide the rationale for data collection at 24 hours from the surgery only. For example, the HEROICS study collected data up to 48 hours post-procedure (https://doi.org/10.1164/rccm.201503-0516OC). Do you have any additional data after 24 hours from the surgery?

9) Table 1 (page 10) is probably unnecessary and could be reported in the text instead.

10) On page 11, line 4 the Authors did mention the usage of the Mann-Whitney U-test for non-normally distributed variables. However, on page 10, line 2 from bottom the Authors did mention that all variables were reported as means and standard deviations. If there are any non-normally distributed variables (for whom the U-test was used), medians and interquartile ranges should be reported instead. Please clarify this point.

11) Page 12, table 2. The row on sex (female) is not necessary and both rows (males and females) could be merged into a single one (M/F, numbers and percentages). Please also specify all units of measurement (height, cm; weight, kg; serum lactate)

12) Page 13, table 3. Please provide additional details on the "time spent on ITU". Is that hours or days? (I would imagine the latter)

13) Page 13, table 4. Please specify the unit of measurement of C-reactive protein

14) Page 14, line 2 from bottom, and page 15, line 1 please amend the word "hyperlactation" (which would refer to breast-feeding)

15) Abstract, conclusions. Please substitute "could prevent postoperative complications" with "could potentially prevent postoperative complications"

6. PLOS authors have the option to publish the peer review history of their article (what does this mean?). If published, this will include your full peer review and any attached files.

Reviewer #1: No

Reviewer #2: No

Reviewer #3: **Yes: **Dr Giorgio Gentile

---

## [Author Response · Author response to Decision Letter 0]

22 Aug 2020

After modifying the manuscript, we checked that the article meets PLOS ONE's style requirements. Please note that due to the reviewers' comments, we have removed Figure 2 from the manuscript and inserted its data in a new table (Table 4). Moreover, we have added more collected variables to the dataset as per the reviewer’s requests: https://figshare.com/s/919bb3545fd431f0bfc6

We have reviewed the characteristics of the only figure in PACE. The English text of the new version of the manuscript was reviewed by American Journal Experts (we attach the new editing certificate).

2. Please ensure that you include a title page within your main document.

We do appreciate that you have a title page document uploaded as a separate file, however, as per our author guidelines (http://journals.plos.org/plosone/s/submission-guidelines#loc-title-page) we do require this to be part of the manuscript file itself and not uploaded separately. Could you therefore please include the title page into the beginning of your manuscript file itself, listing all authors and affiliations.

We have removed the title page file and incorporated it into the manuscript.

3. Thank you for submitting your clinical trial to PLOS ONE and for providing the name of the registry and the registration number. The information in the registry entry suggests that your trial was registered after patient recruitment began. PLOS ONE strongly encourages authors to register all trials before recruiting the first participant in a study.

i) your reasons for your delay in registering this study (after enrolment of participants started);

The record of this study was made after the data had been extracted by the software that collected the variables. The database created by the information collection system did not allow extraction, and we needed the help of a specialist technician to be able to access the data. This is the reason for the delay.

ii) confirmation that all related trials are registered by stating: “The authors confirm that all ongoing and related trials for this drug/intervention are registered”.

Done (design section).

Please also ensure you report the date at which the ethics committee approved the study as well as the complete date range for patient recruitment and follow-up in the Methods section of your manuscript.

The project was approved by the Ethics Committee on December 2nd, 2016. Recruitment began on September 1st, 2017, and ended on February 28th, 2018. We have added this information in the article.

5. Review Comments to the Author

Reviewer #1:

Thank you for your comments and methodological considerations; we have learned from your statistical observations.

The English needs some minor attention throughout (eg. potency should be power, and I think variables of confusion may actually mean confounders).

You are absolutely right. We have reviewed these words.

Given the fact that 64 patients with data were required why was no margin for error built in - there would appear to be about a 15% dropout; it would also appear that these dropouts may be informative given that they are associated with clinical outcome. How was this allowed for in the analysis? Why are no patients excluded from analysis in the CONSORT diagram - 10 are excluded from a full ITT analysis.

The analysis was performed on patients who were not eliminated from the study who had received allocated intervention, as the number of dropouts was equal in both groups (5 and 5). No action was taken since it did not influence the real situation of the two groups.

Patients were randomised into one of 2 arms, not 8 blocks - do you mean the block size in the randomisation was 8? Was there any stratification? It is statistically incorrect to test baseline characteristics by arm. Units are missing from table 2.

The website "http://www.randomization.com/" was used, which divided 64 patients into two groups of 32 and divided these into groups of 8 randomized patients with similar values. There was no stratification. We removed the columns about contrast tests and added the units.

The numbers male and female are wrong as the HG numbers add up to 29 not 27

We apologize; this was a misprint. M=15 and F=12.

It is incorrect to analyse the primary outcome within arm - please perform a proper repeated measures analysis adjusted for baseline. As such the contrast between groups and the CI are not given here- I also assume the GC really means CG.

The data require a proper analysis to give the treatment effect and CI with inferences based upon the CI. Until these are done the value of the intervention is not assessable.

We have performed an intergroup analysis that we have detailed in a new table (table 4). We have also added a paragraph explaining it together with the CI.

GC means CG. We corrected the misprint.

Table 4 uses intervention and control - the rest of the paper uses HG and CG.

We have changed the name of Table 4 (now Table 3).

Reviewer #2:

Thank you for your specific considerations divided into sections. We have been able to detect the problems well enough to be able to solve them.

In my opinion, the biggest flaw is that the main outcome of this trial was the “only” reduction of lactatemia, which it may be interesting per sè, but you should provide further insights to better interpret the findings of the study. Accordingly, you declare in the abstract that the aim of study is “whether continuous ultrafiltration…. decreases post-operative lactatemia and its consequences”.

What are these consequences? You reported the reduction of lactatemia in methods (in the abstract and in the main body) as unique outcome. In the phase of the trial design did you think other secondary outcomes in addition to reduction of blood lactate levels?

It was mainly the reduction in the intubation time. Other secondary outcomes are a reduced length of hospital stay and reduced occurrence of postoperative morbidity and mortality. It also improves liver function and reduces inotropic use time in low-risk patients according to the EuroSCORE.

We have added some information in the article along with bibliographic references in the last paragraphs of the discussion.

Moreover, how do you interpret the reduction of lactatemia? Is it a result of better depuration or a consequence of the general improvement of hemodynamic conditions and/or inflammation? Can you provide further insight to clarify this point?

Furthermore, there is a formal issue, because style of this manuscript is similar to a narrative review (mostly in the introduction and discussion sections). It needs a further effort to improve this point.

There are two reasons: This procedure eliminates significant amounts of inflammatory mediators and improves liver function. We added this information with references in the article.

The first paragraphs of the discussion refer to the explanation of the factors that could influence high-volume hemofiltration in our study; therefore, at first, it seems like a narrative review. Despite this, we improved the interpretation of our findings in the second part of the discussion.

Following my point-by-point considerations:

- Abstract:

o Please clarify the points described above, changing it accordingly.

We have incorporated some of the information you requested without exceeding the word limit indicated by PLOS ONE in the abstract.

- Introduction section:

o Please reduce this section, providing only the information tightly related to the topic of the manuscript (e.g.: please cancel the paragraph about use of type of solutions, etc.).

We have reduced the length of some paragraphs, especially those that offered some more general information. We followed your suggestion, and we deleted the paragraph about the type of solutions.

- Methods section:

o Please cancel table 1 and move the information from the table to text.

o Please add secondary endpoints, if it is possible.

We deleted table 1 and added all the information to the text.

As secondary endpoints, kidney function and the use of inotropes were analyzed, but the results obtained indicate that a larger sample of patients is needed.

- Results section:

o Please add information about patients’ hemodynamics and other vital parameters for improving the interpretation of findings.

We added a subsection in the results. We studied other variables (attendance time between unclamping of the aorta and the completion of ECC, time of use of inotropic agents) that affect patients’ hemodynamics.

o Please add in the table 3 the value at baseline and the intragroup differences (pre- and – post continuous UF).

The lactate values and intragroup differences are provided in Table 4 (now Table 3). Intergroup differences are listed in Table 4.

o Please use regression analysis to analyze the effect after correction for confounders (ECC time, etc.)

All R2 values are reported in Table 5, but we have added several paragraphs about their analysis before the table.

o Please clarify the legend and the parameters included in the table 4.

We have clarified the parameters and the legend more precisely in Table 4 (now Table 3).

- Discussion section:

o Depending on these additional results, please reconsider the discussion section and the limitations of the study.

o Please rewrite discussion, highlighting the main findings of study and focused on the interpretation of these findings.

We added another limitation and more interpretation of our findings, especially in the second part of the discussion.

Reviewer #3:

Thank you for your point of view and for your specific comments.

1) Although I do appreciate that the topic is complex and requires some background information, the introduction is probably a bit too long and could be slightly shortened

We have reduced the length of some paragraphs, especially those that offered some more general information.

2) As there is no universal consensus on the definition of high-volume hemofiltration, the Authors should provide additional information on the rate of hemofiltration. Based on the results of two large randomized-controlled trials (N Engl J Med 2009, 361:1627–1638; N Engl J Med 2008, 359:7–20) and subsequent systematic reviews (Crit Care Med 2010, 38:1360–1369;Clin J Am Soc Nephrol 2010, 5:956–963)3, the adequate dose of hemofiltration treatment for acute kidney injury ) has been defined as an effluent rate between 25 and 30 ml/kg/hour, which is consistent with HVHF being defined by an effluent rate exceeding 35 ml/kg/hour. How was the hemofiltration rate in this trial? On page 9, line 11 the Authors have mentioned some information on the blood flow (between 100 and 500 mL/min depending on the time of the surgery), but I could not find any information on the average effluent rate. Please provide this information.

The effluent rate was 110 ml/min, and the average was 80 ml/kg/h, which is similar to the rate in the HEROICS study; however, our study was performed during cardiac surgery with ECC, while the HEROICS study was performed during the postoperative period. We have added these data in the article.

3) Page 6, design. Please specify the name of the hospital in Andalusia. Similarly, in the ethics section on page 7, please provide the name of the hospital where the ethics committee was based.

We have specified the information requested.

4) Page 7, line 3 from bottom. Please provide additional details on the "hemophilia technique". Was this a typo for hemofiltration?

Thank you for observing that mistake. It was a typo for hemofiltration.

5) Page 8, line 1. Please provide additional details on the allocator and the randomization technique. Who was the allocator? Was he one of the Authors or an external allocator? How were the patient assigned to the eight different blocks?

The allocator was assigned by the head of the hospital's ethics committee; thus, it was an external allocator using the website http://www.randomization.com/. The procedure divided 64 patients into two groups of 32, and these were divided into groups of 8 randomized patients with similar values. There was no stratification.

6) Page 8, line 7 from bottom. Please provide the full definition of the acronym "PVC".

We added its full name.

7) Page 9, line 9. Please provide a supporting reference got the analyzer which was used in the study (including the name of the manifacturer).

The names of the manufacturers can be found in the article. We have added the following references:

- Ottens J, Baker RA, Newland RF, Mazzone A. The future of the perfusion record: automated data collection vs. manual recording. J Extra Corpor Technol. 2005;37: 355–359.

- LivaNova. Connect system. 2017 [cited 12 August 2020]. In: Connect [Internet]. London: LivaNova - . [about 3 screens]. Available from: https://www.livanova.com/en-US/Home/Products-Therapies/Cardiovascular/Healthcare-Professionals/Cardiopulmonary/Data-Management-Systems/Connect.aspx.

8) Page 9, line 2 from bottom. Please provide the rationale for data collection at 24 hours from the surgery only. For example, the HEROICS study collected data up to 48 hours post-procedure (https://doi.org/10.1164/rccm.201503-0516OC). Do you have any additional data after 24 hours from the surgery?

Our response is similar to the one we discussed earlier. The HEROICS study was conducted postoperatively in cardiac surgery patients with severe shock who required high doses of catecholamines (epinephrine>0.2 μg/kg/min, noradrenaline>0.4 μg/kg/min, or epinephrine + [norepinephrine/2]>0.2 μg/kg/min) or cardiovascular assistance through extracorporeal membrane oxygenation/extracorporeal life support (ECMO) (within 3 to 24 hours of admission to the intensive care unit.

High-volume hemofiltration was performed for 48 continuous hours at a rate similar to that used in our study (80 ml/kg/h).

9) Table 1 (page 10) is probably unnecessary and could be reported in the text instead.

We have moved the information from the table to the text.

10) On page 11, line 4 the Authors did mention the usage of the Mann-Whitney U-test for non-normally distributed variables. However, on page 10, line 2 from bottom the Authors did mention that all variables were reported as means and standard deviations. If there are any non-normally distributed variables (for whom the U-test was used), medians and interquartile ranges should be reported instead. Please clarify this point.

As a result of the suggestion for modification by another reviewer, we have included a Mann-Whitney U test (with its median and interquartile ranges). Therefore, we have clarified this in the methodology.

11) Page 12, table 2. The row on sex (female) is not necessary and both rows (males and females) could be merged into a single one (M/F, numbers and percentages). Please also specify all units of measurement (height, cm; weight, kg; serum lactate)

We have changed the table according to your observations.

12) Page 13, table 3. Please provide additional details on the "time spent on ITU". Is that hours or days? (I would imagine the latter)

Yes (days). We have added all units in the table.

13) Page 13, table 4. Please specify the unit of measurement of C-reactive protein

As mentioned above, its unit of measurement has also been included.

14) Page 14, line 2 from bottom, and page 15, line 1 please amend the word "hyperlactation" (which would refer to breast-feeding)

We have changed the term to other expressions.

15) Abstract, conclusions. Please substitute "could prevent postoperative complications" with "could potentially prevent postoperative complications"

We added this word.

---

## [Decision Letter · Decision Letter 1]

5 Oct 2020

PONE-D-20-10699R1

Continuous ultrafiltration during extracorporeal circulation and its effect on lactatemia: A randomized controlled trial

PLOS ONE

Dear Dr. Marín-Paz,

Thank you for submitting your manuscript to PLOS ONE. After careful consideration, we feel that it has merit but does not fully meet PLOS ONE’s publication criteria as it currently stands.

All referees found you manuscript substantially improved but a few minor issues deserve your attention. Therefore, we invite you to submit a revised version of the manuscript that addresses the minor points raised by reviewer #2.

We look forward to receiving your revised manuscript.

Kind regards,

Gianpaolo Reboldi, MD, MSc, PhD

Academic Editor

PLOS ONE

Reviewers' comments:

Reviewer's Responses to Questions

**Comments to the Author**

1. If the authors have adequately addressed your comments raised in a previous round of review and you feel that this manuscript is now acceptable for publication, you may indicate that here to bypass the “Comments to the Author” section, enter your conflict of interest statement in the “Confidential to Editor” section, and submit your "Accept" recommendation.

Reviewer #1: All comments have been addressed

Reviewer #2: All comments have been addressed

Reviewer #3: All comments have been addressed

2. Is the manuscript technically sound, and do the data support the conclusions?

Reviewer #1: (No Response)

Reviewer #2: Yes

Reviewer #3: Yes

3. Has the statistical analysis been performed appropriately and rigorously? 

Reviewer #1: (No Response)

Reviewer #2: Yes

Reviewer #3: Yes

4. Have the authors made all data underlying the findings in their manuscript fully available?

Reviewer #1: (No Response)

Reviewer #2: (No Response)

Reviewer #3: Yes

5. Is the manuscript presented in an intelligible fashion and written in standard English?

Reviewer #1: (No Response)

Reviewer #2: Yes

Reviewer #3: Yes

6. Review Comments to the Author

Reviewer #1: (No Response)

Reviewer #2: The authors have significantly improved the manuscript. However, it needs some minor revisions:

- Abstract:

o Please declare secondary endpoints (Reduction in the intubation time, etc.)

- Methods section:

o Please declare secondary endpoints (Reduction in the intubation time, etc.)

- Results section:

o I think that there is typo in table 2 (Euroscore: 94.5±72.40 in low risk CG group)

I advise the authors to make an extra effort to reduce the introduction and improve the discussion

Reviewer #3: The authors have satisfactorily addressed my previous comments. Hence, in my opinion the manuscript is now suitable for publication in Plos One.

7. PLOS authors have the option to publish the peer review history of their article (what does this mean?). If published, this will include your full peer review and any attached files.

Reviewer #1: No

Reviewer #2: No

Reviewer #3: **Yes: **Dr Giorguo Gentile

---

## [Author Response · Author response to Decision Letter 1]

19 Oct 2020

Reviewer #2: The authors have significantly improved the manuscript. However, it needs some minor revisions:

- Abstract:

o Please declare secondary endpoints (Reduction in the intubation time, etc.)

We have added the main secondary endpoints in the study (reduction in the intubation time and time spent in ICU). We have reduced the introduction section of the abstract in order not to exceed the word limit.

- Methods section:

o Please declare secondary endpoints (Reduction in the intubation time, etc.)

We have added the secondary endpoints in the study (possible benefits in patients with renal failure and the reduction in the intubation time, time spent in ICU and C-reactive protein levels).

- Results section:

o I think that there is typo in table 2 (Euroscore: 94.5±72.40 in low risk CG group)

There is no typo. This result with high variability is due to the low number of low-risk patients. We have highlighted this information in the discussion.

I advise the authors to make an extra effort to reduce the introduction and improve the discussion

We have reduced the length of the introduction. If you feel that more changes should be made in the introduction and discussion, we would appreciate it if you could point out the paragraphs that need to be modified.

Thank you for your time and considerations.

---

## [Editor Report · Decision Letter 2]

3 Nov 2020

Continuous ultrafiltration during extracorporeal circulation and its effect on lactatemia: A randomized controlled trial

PONE-D-20-10699R2

Dear Dr. Marín-Paz,

We’re pleased to inform you that your manuscript has been judged scientifically suitable for publication and will be formally accepted for publication once it meets all outstanding technical requirements.

Kind regards,

Gianpaolo Reboldi, MD, MSc, PhD

Academic Editor

PLOS ONE

---

## [Editor Report · Acceptance letter]

12 Nov 2020

PONE-D-20-10699R2 

Continuous ultrafiltration during extracorporeal circulation and its effect on lactatemia: A randomized controlled trial 

Dear Dr. Marín-Paz:

I'm pleased to inform you that your manuscript has been deemed suitable for publication in PLOS ONE. Congratulations! Your manuscript is now with our production department. 

Kind regards, 

on behalf of

Prof Gianpaolo Reboldi 

Academic Editor

PLOS ONE